# Micro- and Nanofluidic pH Sensors Based on Electrodiffusioosmosis

**DOI:** 10.3390/mi15060698

**Published:** 2024-05-25

**Authors:** Tadashi Takagi, Tatsunori Kishimoto, Kentaro Doi

**Affiliations:** Department of Mechanical Engineering, Toyohashi University of Technology, Toyohashi 441-8580, Aichi, Japankishimoto@me.tut.ac.jp (T.K.)

**Keywords:** nanochannel, ionic current rectification, ion selectivity, electrodiffusioosmosis, pH sensor

## Abstract

Recently, various kinds of micro- and nanofluidic functional devices have been proposed, where a large surface-to-volume ratio often plays an important role in nanoscale ion transport phenomena. Ionic current analysis methods for ions, molecules, nanoparticles, and biological cells have attracted significant attention. In this study, focusing on ionic current rectification (ICR) caused by the separation of cation and anion transport in nanochannels, we successfully induce electrodiffusioosmosis with concentration differences between protons separated by nanochannels. The proton concentration in sample solutions is quantitatively evaluated in the range from pH 1.68 to 10.01 with a slope of 243 mV/pH at a galvanostatic current of 3 nA. Herein, three types of micro- and nanochannels are proposed to improve the stability and measurement accuracy of the current–voltage characteristics, and the ICR effects on pH analysis are evaluated. It is found that a nanochannel filled with polyethylene glycol exhibits increased impedance and an improved ICR ratio. The present principle is expected to be applicable to various types of ions.

## 1. Introduction

Micro- and nanofluidic channel devices have attracted much attention for applications such as micromixers [1], microreactors [2,3], and biosensors [4,5] because sufficient sensitivity can be achieved for small samples. Furthermore, nanoscale test sections have a large surface-to-volume ratio that can modify the properties of liquids, causing counterions of surface charges to be dominant. It was reported that ionic current rectification (ICR) was attained by controlling ion transport through asymmetric channels such as glass capillaries [6] and polymer nanopores [7] or by using ion concentration differences [8]. Cheng and Guo [9] demonstrated that for a pair of KCl baths with different concentrations connected by a nanochannel, where the ion concentration was fixed by the surface charge density, ionic current rectification occurred depending on the direction of applied electric fields. Asymmetric provision of additional ions from the baths resulted in asymmetric current–voltage (*I*–*V*) characteristics. Especially for dilute solutions, electric double layers (EDLs) induced by the surface charges became thicker and bound the number of counterions [10,11]. Such conditions induce an electroosmotic flow (EOF) [12] or an electrohydrodynamic (EHD) flow [13] under an externally applied electric field. Although ion selectivity is often provided using ion exchange membranes [14,15], ionophores [16,17], or glass nanopores [18,19,20], recent fabrication techniques enable us to artificially design nanoscale structures that achieve ion selectivity. Using EOF and EHD flows, the velocity control of biomacromolecules, e.g., deoxyribonucleic acid (DNA) and ribonucleic acid (RNA) molecules, was also proposed [21,22,23]. Furthermore, chemically modified channel surfaces and their junctions were used to produce an ionic diode [24,25] and an ion-sensitive field-effect transistor [26]. Cai et al. [18] successfully modified the surfaces of glass pipette tips to detect DNA by ICR measurements. Ma et al. [19] developed an aptamer-functionalized nanopipette for the detection of N protein, which was effective for the identification of SARS-CoV-2. N protein adsorption was recognized by changes in the ICR response. Lin et al. [27] numerically simulated nanoparticle detection using bipolar conical nanopores and found that the combination of bipolar conical nanopores and the ICR effect improved the capture frequency for target particles and the measurement accuracy. Several numerical analyses suggested that the ICR behavior of asymmetric nanochannels effectively responded to the solution pH [28] and heat conduction [29]. Applications of ion concentration polarization and ICR for micro-nanofluidic systems have been summarized in a review article [30]. As described above, ICR effects induced by the large surface-to-volume ratio and asymmetric channel geometries strongly influence nanoscale ion transport phenomena.

We have also investigated ICR effects on ion selectivity in micro- and nanofluidic channels, which are applied to EHD flows through ion-exchange membranes [31,32] and proton concentration measurements [33,34]. The results suggested that EOF and EHD flows could be induced with quite low ion concentrations [13]. The ion selectivity is possibly improved as the ion concentration decreases [13,33,34] because EDLs in nanochannels tend to overlap between opposite surfaces at quite low concentrations with electrical charge carriers limited to surface charges [11]. Furthermore, ion concentration gradients in a nanochannel cause diffusion in ionic currents along the channel [8,34]. That is, the resistivity increases when both the electrophoresis and electroosmosis of ions are in the counter direction to the concentration diffusion direction, an effect that is termed electrodiffusioosmosis [8,34]. Glass pipettes or glass films are usually used for electrodes for pH measurements because of the proton selectivity and high conductivity of glass nanopores [35] due to a unique proton transport mechanism in water [36,37]. Recent developments have overcome the shortcomings of conventional equilibrium state measurements, which are limited to a slope of 59.1 mV/pH at 298 K, which is referred to as the Nernst limit [38].

In this study, pH sensors with different shapes are fabricated on thermally oxidized silicon substrates (Si/SiO_2_) and operated based on ICR and electrodiffusioosmosis. Our goal is schematically shown in Figure 1. Previously demonstrated glass microelectrodes, which were portable and easy to move to the measurement points, were fabricated based on the same principle [33,34]. On the other hand, in the present method, the electrodes are stationary and can be embedded in micro- and nanochannels. The remainder of this paper is organized as follows. The fabrication process for the micro- and nanofluidic channels and the experimental methods are described in the Materials and Methods section. The experimentally determined *I*–*V* characteristics in the presence of pH differences across the nanochannels are discussed in the Results and Discussion section. Finally, new findings and perspectives are summarized in the Conclusions section.

## 2. Materials and Methods

### 2.1. Nanochannels for the Test Section

In this study, the proton concentration in sample solutions was sensed by the difference in electrical conductivity in the forward and backward directions of the ionic current. To separate the sample and reference solutions, nanochannels were used. The composite structure of micro- and nanochannels is effective in focusing strong electric fields on the nanochannel, which serves as the sensing region [39]. Three types of nanochannels were prepared for pH sensors, and their cross-sectional views are depicted in Figure 2. The nanochannels and reservoirs were fabricated on Si/SiO_2_ substrates. Figure 2a shows a conceptual diagram of a nanochannel, hereafter referred to as Type A. A nanochannel with a cross section of 500×500 nm^2^ and a length of 20 µm was sealed with a PDMS substrate and acted as the test section. The working electrode (WE), counter electrode (CE), and reference electrode (RE) were set in different reservoirs. Figure 2b shows another type of nanochannel, hereafter referred to as Type B. Nanochannels with a cross section of 500×500 nm^2^ and a length of 475 µm were separately connected to each reservoir. The WE and CE were placed in the separate reservoirs, and the RE wes placed in the inlet of the microchannel, as described later. Figure 2c shows the third type of nanochannel, hereafter referred to as Type C. In this case, three nanochannels with reservoirs were formed on the Si/SiO_2_ substrate, and the WE, CE, and RE were placed in three of the four reservoirs. A sample solution was poured into the central reservoir. The nanochannels had a cross section of 500×500 nm^2^ and a length of 20 µm. In the Type C structure, empty reservoirs were prepared for future multi-electrode measurements; three were used for electrodes, and the central one was for samples. Three-dimensional views of the Type A, B, and C structures are depicted in Figure 2d–f, respectively. For each type of channel, the test section, which connects the sample and reference solutions, was designed with the same cross section of 500×500 nm^2^ and the length of 20 µm for the Type A and C and 475 µm for the Type B structures, although the microchannels for introducing liquids varied depending on the channel types, as shown in the focused view in Figure 2g–i, respectively.

For the Type A structure, both the RE and CE were placed in the reference solution (a mixture of pH 1.68 buffer and KCl solutions), and the WE wes placed in the sample solution. This is a simple two-electrode, two-solution method. In the Type B structure, the three electrodes were isolated from each other; the RE was in the sample solution, the WE was in the KCl solution, and the CE was in the reference solution. In the Type C structure, the sample solution was fully isolated from the three electrodes and was connected to them via nanochannels, where the WE and RE were in the KCl solution and the CE was in the reference solution.

Each type of nanochannel was fabricated on a four-inch Si/SiO_2_ substrate with a thermal oxide thickness of about 700 nm. The nanochannels and reservoirs were patterned in the SiO_2_ layer using a photolithography technique, as depicted in Figure 3. A silane coupling agent (hexamethyldisilazane, HMDS) and a positive photoresist (THMR–iP3000, Tokyo Ohka Kogyo, Kawasaki, Japan) were coated on the SiO_2_ surface using a spin coater with a rotation speed of 4000 rpm. After pre-baking at 383 K for 120 s using a hotplate, nanochannel patterns were formed on the SiO_2_ surface using a photomask and an i-line stepper (NSR–THFi16CH, Nikon Tec, Tokyo, Japan). The exposure time depended on the channel type. After a post-exposure bake at 383 K for 90 s, the substrate was immersed in an NMD–3 developing agent. The developing time was 30 to 40 s depending on the channel type. Finally, the photoresist was hardened on a hotplate at 383 K for 300 s. Next, the channel pattern on the SiO_2_ surface was processed using reactive ion etching (RIE, NR-10, Samco, Kyoto, Japan). The etching time was optimized for an etching depth of 500 nm. Finally, residues were removed by oxygen plasma ashing for 10 min.

### 2.2. Microchannels for Sample Transport

The microchannels that seal the upper face of nanochannels were made from PDMS (Sylgard 184, Dow Silicones, Midland, MI, USA) using a soft lithography technique. A negative photoresist (SU–8 3050, Kayaku Advanced Materials, Westborough, MA, USA) was coated on a Si wafer at a rotation speed of 1800 rpm for 60 s using a spin coater (MS–B100, Mikasa, Tokyo, Japan), and the substrate was baked on a hotplate at 338 K for 2 min and 368 K for 5 min. The microchannel patterns were printed on the SU–8 surface with a maskless UV exposure system (µMLA, Heidelberg Instruments, Heidelberg, Germany) and baked at 338 K for 1 min and 368 K for 6 min. The patterned substrate was then immersed in a developer (SU–8 Developer, Kayaku Advanced Materials, Westborough, MA, USA) for 8 min and rinsed in isopropanol for 60 s. Finally, the substrate was baked at 423 K for 20 min. Liquid PDMS was poured onto the SU–8 mold fixed in an aluminum vessel and baked at 373 K for 2 h. The hardened PDMS substrate was peeled off from the microchannel mold. An approximately 20×20×10 mm^3^ (W×L×H) PDMS substrate on which microchannels were printed was cut off using a medical scalpel, and 1.5 mm (Type C) or 2 mm (Type A and B) diameter through-holes for reservoirs and liquid injection were made using a biopsy punch.

Three types of microchannels were designed for nanochannels, Type A, B, and C, as shown in Figure 2. The surfaces of the nanochannels and microchannels were exposed to air plasma for 15 s (SEDE, Meiwafosis, Tokyo, Japan) and were bonded together, being aligned using a microscope. In the Type A structure, as shown in Figure 2d, there was a 20 µm gap at the center between the two microchannels to seal the upper face of the nanochannel. Liquids injected from the through-holes of the PDMS substrate were provided into the nanochannel through the connected microchannels. The width and height of microchannels were designed to be 50 µm each.

### 2.3. Preparation of Sample Solutions

Sample solutions were prepared by mixing a standard pH solution and a KCl aqueous solution. In this study, to verify the linearity of the electrical potential response to the pH difference, aqueous solutions whose pH values are known were used for the samples. For the pH sample solutions, oxalate (pH 1.68), phthalate (pH 4.01), phosphate (pH 6.86), tetraborate (pH 9.18), and carbonate (pH 10.01) standard solutions (Kanto Chemical, Tokyo, Japan) were used, and more details of the compositions are summarized in Table 1. For the measurements, these standard pH solutions were mixed in a 1:1 volume ratio with a 10 mmol/L KCl solution to generate ionic currents using Ag/AgCl electrodes. The pH values of the prepared sample solutions were confirmed in advance using a conventional pH meter (LAQUAtwin, Horiba, Kyoto, Japan). However, the volume of the sample solution poured into the reservoir was only approximately 0.2 mL, and for such a small volume, it is difficult to measure the pH after the experiment. Therefore, standard pH solutions were used to maintain the pH value during the measurements as a first step.

### 2.4. Experimental Methods

The experimental setup is shown in Figure 4. A micro- and nanofluidic channel chip was set on a hotplate, and the WE, CE, and RE of the galvanostat (Versastat 4, Ametek, Berwyn, PA, USA) were placed in the through-holes in the PDMS substrate. The electrical potential difference between the WE and RE was measured while maintaining a galvanostatic current between the electrodes. The current was kept constant for 20 s (Type A and B) or 60 s (Type C) to measure the steady-state potential difference and was then increased in a stepwise manner. The current direction was inverted by changing the polarization of the electrodes using a mechanical switch. The resolution of the applied current was 20 pA, and the measurement frequency was set to 10 Hz. The steady-state potential difference was averaged over the measurement time with a 95% confidence interval. During the measurement, a micro- and nanofluidic channel was fixed to a hotplate set at 308 K by applying thermal grease on the back side. In the Type A structure, as shown in Figure 2d, a mixed solution of 10 mM KCl and pH 1.68 buffer, which is the reference solution, was poured into the CE and RE reservoirs, and the sample solution with 10 mM KCl was placed in the WE reservoir. Both reservoirs were connected by a nanochannel in the SiO_2_ layer. Two-electrode measurements were then performed. In the Type B structure, as shown in Figure 2e, the reference and sample solutions were poured into separate reservoirs in the PDMS substrate in the same way as for the Type A structure. However, the RE was separated from the CE and was set in the microchannel that bridged the two nanochannels connected to the WE and CE reservoirs. In the Type C structure, as shown in Figure 2f, the sample solution was poured into the central reservoir in the PDMS substrate. The WE, CE, and RE were set in reservoirs that were connected to the sample solutions via nanochannels in the SiO_2_ layer. In this structure, some reservoirs were left empty and were reserved for future multi-electrode measurements. Three-electrode measurements were performed for the Type B and C structures. The applied current was optimized for each channel type to measure the potential difference within an appropriate potential range. The galvanostat served as a power source and amplifier to maintain constant current conditions. A Ag–AgCl wire, which was made from a Ag wire with a diameter of 0.3 mm soaked in a sodium hypochlorite aqueous solution until the surface color changed from silver to black, was used for each electrode. The WE and CE served as the anode and cathode, respectively, and an ionic current was generated due to the reactions Ag + Cl^−^→ AgCl + e^−^ at the anode and AgCl + e^−^→ Ag + Cl^−^ at the cathode. In the experimental setup, as shown in Figure 4, a Ag–AgCl wire, used for the WE, CE, and RE, was soldered with a lead connected to the galvanostat via the mechanical switch and was fixed on the manipulator arm to soak it in the liquid in a reservoir in the PDMS substrate.

The liquid reservoirs made by through-holes in the PDMS substrate were approximately 10 mm high and were fully filled with solutions. The solutions were injected into each reservoir using syringes, and the micro- and nanochannels were degassed in a vacuum chamber to fill the nanochannels with solutions. Whether the channel was filled with liquid could be confirmed optically using a microscope and electrically by conduction. Furthermore, by additionally injecting the solutions, the reservoirs were fully filled with liquids. In particular, sample solutions were sequentially replaced by the next samples with a lower pH. In the sequence, a sample solution was replaced after co-washing using the next sample. In the present experiments, the head of each reservoir was not controlled, and the effect on the *I*–*V* characteristics remain unknown. Although the visualization of nanochannels will be reported in the near future, this study empirically assumes that small head differences do not significantly affect the experimental results.

## 3. Results and Discussion

Firstly, the *I*–*V* characteristics of micro- and nanochannel chips were investigated. Typical results for the Type A structure are shown in Figure 5. In this case, a mixture of 10 mM KCl and pH 1.68 buffer solutions at a volume ratio of 1:1 was filled in both reservoirs, and the electrical potential response was measured by applying galvanostatic currents. The electrical potential responses to ionic currents in positive and negative directions were measured. It was found that the electrical potential converged to a steady state after the transient response, although figures are omitted here. Figure 5 shows the steady-state *I*–*V* characteristics for each current condition. The slope (resistance) is constant at 85.5 MΩ and is unique regardless of the current direction. That is, the micro- and nanochannels and the electrodes symmetrically respond to the ionic currents with no concentration difference between the two liquid reservoirs. In the Type B and C structures, similar trends were confirmed, although the figures are omitted here.

### 3.1. Electrical Responses of Type A Micro- and Nanofluidic Channels

Using the Type A structure, the electrical responses and *I*–*V* characteristics were investigated. Figure 6a,b shows the electrical potential responses for various pH sample solutions as a function of time and presents the results for galvanostatic currents in the positive and negative directions, respectively. Here, the CE and RE were initially placed in a reservoir in which a mixture of 10 mM KCl and pH 1.68 solutions at a volume ratio of 1:1 was poured, and the WE was placed in another reservoir with a mixture of 10 mM KCl and sample solutions at a volume ratio of 1:1. In this situation, we defined the current direction from the WE to the CE and RE as positive, and the electrode positions were exchanged using a mechanical switch to apply a negative current. As shown in Figure 6a, the electrical potential increased with an increasing galvanostatic current, such as 5.0, 10.0, and 15.0 nA. Furthermore, the potential difference seemed to increase with an increasing pH. To evaluate the steady states, the electrical potentials were averaged over the last 10 s at each current level that was maintained for 30 s. Figure 6b shows the electrical potential responses to negative currents for samples with various pH values. The electrical potential increased with increasing galvanostatic current in the negative direction. The time-averaged potential differences are presented as a function of galvanostatic current in Figure 6c. Each data point and error bar are the results of five experiments. The slope of the potential differences shows linearity for both current directions, and the conductivity varied for pH sample solutions. We focus on the positive currents in which the ionic current was directed from the sample to the reference solutions, that is, from a solution with a lower proton concentration to a solution with a higher one. For acid solutions (pH < 7), the slope became larger with increasing pH values of the sample solution. This result implies that the difference in proton concentrations is reflected in the slope due to the proton selectivity of the nanochannel. However, alkaline sample solutions resulted in smaller slopes than acid samples. Concentrations of the other electrolyte ions possibly contributed to the conductivity, although the proton concentration became lower with increasing pH value. Additionally, for negative currents, there was no clear trend regarding the pH values of the samples.

### 3.2. Electrical Responses of the Type B Structure

For the micro- and nanochannels in the Type B structure, the electrical potential responses for galvanostatic currents were measured in the same way as for the Type A structure. In the Type B structure, the RE separated from the CE was immersed in a microchannel across the nanochannels connected to the WE and CE reservoirs. The CE was placed in a mixture solution of 1 mM KCl and a 10-fold dilution of pH 1.68 buffer at a volume ratio of 1:1. The WE was placed in a 1 mM KCl solution, and the RE was placed in a mixture of 1 mM KCl and sample solutions at a volume ratio of 1:1. Figure 7a,b shows the electrical potential responses for sample solutions with various pH values to positive and negative currents, respectively. The electrical potentials were measured for applied galvanostatic currents of 0.1, 0.2, and 0.3 nA and were averaged using data points for the last 10 s within 30 s measurements. For the positive currents, the electrical potential increased as the applied current increased, as shown in Figure 7a. On the other hand, the electrical potential responses changed for the negative currents, as shown in Figure 7b. The *I*–*V* characteristics for the pH sample solutions are summarized in Figure 7c. Focusing on the positive currents, the slope of the electrical potential tends to become larger as the pH of sample solution increases. The slopes for alkaline solutions of pH 9.18 and 10.01 were larger than those for acid solutions. This result indicates that the proton selectivity of the nanochannel between the RE and CE reservoirs worked well and that transport of the other ions was effectively blocked in the nanochannel. However, sample solutions with pH greater than 9 may not be identified with sufficient sensitivity because the slope for pH 10.01 is estimated to be smaller than that for pH 9.18. Focusing on the negative currents, the slopes seem to vary widely. Due to the weak currents applied, the potential responses became extremely small. Furthermore, the electrophoretic transport and concentration diffusion of protons were in the same direction for the negative current conditions. When concentration diffusion contributed more dominantly to the ionic current than electrophoretic transport, the electric field was possibly inverted. Ion transport phenomena driven by electrokinetic transport and concentration diffusion involving liquid flows are known as electrodiffusioosmosis [8]. We have also examined electrodiffusioosmosis associated with a pH sensor using glass microelectrodes while proposing a theoretical model [33,34]. The fraction of electrophoresis, electroosmosis, and diffusion in ionic currents is reflected in the electrical conductivity, and the ion selectivity of nanochannels causes ICR [6,40,41,42,43]. The present system may require more improvement and optimization to clearly provide ICR effects.

### 3.3. Electrical Responses of the Type C Structure

To improve the ICR ratio, we modified the design of the micro- and nanochannels. In the Type C structure, as shown in Figure 2f, the reservoir of sample solutions was separated from the reservoirs for the WE, CE, and RE, which were connected to the sample solution via nanochannels. Compared with the Type B structure, the RE, which was immersed in a 10 mM KCl solution, was also separated from the sample solution. The CE was placed in a mixture of 10 mM KCl and pH 1.68 buffer solutions at a volume ratio of 1:1, and the WE was placed in a 10 mM KCl solution. Only the sample solution with pH 6.86 was diluted 10 times because the electrical conductivity was much higher than that for the other solutions. In a previous study [33,34], we suggested that narrow channels filled with agarose gel or polyethylene glycol (PEG) increase the impedance and improve the ICR ratio. Based on this, a mixture of PEG and 10 mM KCl solution was poured into a nanochannel between the reservoirs for the CE and the sample solution before injecting the reference and sample solutions. Applying 1.0, 2.0, and 3.0 nA in positive and negative directions between the WE and CE, the electrical potentials were measured, as shown, respectively, in Figure 8a,b. In this case, the electrical potentials stably converged to constant values in both directions at each current level. In Figure 8b, the electrical potentials are usually small in the negative direction, although an extremely high response is observed from a sample solution of pH 10.01. Figure 8c summarizes the *I*–*V* characteristics resulting from Figure 8a,b. Except for the case of pH 10.01, the slopes of the potential differences were larger for the positive currents than for the negative currents. This result indicates that the ionic current was clearly rectified by the concentration difference between the sample and reference solutions via the nanochannel. Focusing on the positive current, alkaline sample solutions were also clearly distinguished from the acid samples. This result also indicates that the separation of each electrode and sample solutions is highly effective for identifying pH differences. Furthermore, it was found that relatively small slopes in the negative currents, except for the case of pH 10.01, are explained by the ICR effect. The nanochannel filled with PEG and KCl solutions between CE and sample solutions worked well to rectify the ionic currents between the positive and negative directions. In the case of pH 10.01, since the proton concentration in the sample solution was very low, it might affect the conductivity of the entire system. In the positive direction, protons in a sample solution with a lower concentration than in the CE reservoir were transported to the CE end through the nanochannel. In this direction, the proton current, which countered the proton diffusion between the sample and CE reservoirs, required electrical potentials higher than the case that both proton current and diffusion were in the negative direction. That is, the proton concentration was well identified from other electrolyte ions included in the sample solutions. The present results showed similar trends as those obtained using glass microelectrodes in previous studies [33,34]. Detailed phenomena associated with the ICR effect and proton concentration identification are explained based on electrodiffusioosmosis, as described in the literature [34].

### 3.4. Theoretical Model of ICR

Figure 9 shows examples of *I*–*V* characteristics and the electrical resistance as a function of ΔpH resulting from the theoretical model presented in a previous study [34] and Appendix A. In Figure 9a, *V* is linearly proportional to *I* based on Ohm’s law. Here, *R* is evaluated using Equations (Equation 5) and (Equation 6) for positive and negative currents, respectively, corresponding to the experimental conditions. We then compare the results for $\tilde{\xi }=0.2$ and 0.5 with α=200 MΩ. With increasing $\tilde{\xi }$, *V* tends to become greater. As shown in Figure 9b, *R* increases with increasing ΔpH, and the slope becomes steeper with $\tilde{\xi }$. These results indicate that the proton conductivity influences the value of $\tilde{\xi }$ that represents the activity of protons compared to the other ions. Therefore, a higher potential difference is required to suppress proton transport maintaining a constant current, as schematically explained in Figure 9c. ΔpH, which causes the degree of diffusion, determines the potential difference to suppress the diffusion current. For negative and positive currents with ΔpH=0, the *I*–*V* characteristics were evaluated using R=200 MΩ. As shown in Figure 9b, the slopes were evaluated to be 40 and 100 MΩ/pH for $\tilde{\xi }=0.2$ and 0.5, respectively. For a galvanostatic current of 1 nA, these slopes can be translated to 120 and 300 mV/pH, respectively. This means that enhanced proton transport relative to other ions requires an electrical potential difference to suppress proton diffusion to maintain a steady current condition, as depicted in Figure 9c.

### 3.5. Comparison between Channel Types

As described above, the proton selectivity of SiO_2_ nanochannels causes rectification of the ionic current, and as a result, the proton concentration was successfully identified based on the effect of electrodiffusioosmosis. The measurement stability and the linearity as a function of pH depended on the structure of the micro- and nanochannels and the configuration of the electrodes. Figure 10a shows the resistance of the Type A, B, and C channels as a function of pH resulting from the positive current domains. Each data point contains N=5, 1, and 13 (except for N=3 for pH 6.86 and N=7 for pH 10.01) experiments using the Type A, B, and C structures, respectively. Solid lines in the figure represent linear fits by the theoretical model described in the previous section and Appendix A. The slopes are 32.6, 278, and 82.3 MΩ/pH for Type A, B, and C, respectively. Each channel type shows close linearity in the acidic domain. Furthermore, the Type B and C structures show good responses even in the alkaline domain. The sensitivity of the pH measurements and correlation coefficient ($R^{2}$) for the linear fits are also shown in Figure 10a. As shown in Figure 10b, the resistance is converted to the potential difference by multiplying the maximum current value used in the experiment. It was found that the slope for the Type C structure is 243 mV/pH at +3 nA and is the highest among the three types. The Type A structure also shows a good response for pH < 7, but tends to be weak for alkaline samples. Although the Type B structure shows good linearity over a wide pH range, the measurement stability should be improved for further applications. The problems with the Type A and B structures were improved using the Type C structure, where each electrode and sample were fully separated and the nanochannel was filled with PEG solution to increase the impedance. Although hydrodynamic effects on the measurement accuracy have not been revealed in this study, it is clear that the high resistivity of nanochannels is effective, as demonstrated by the Type C structure. In addition to the glass microelectrode pH sensor reported in a previous study [34], the effectiveness of our nanochannel pH sensor has also been demonstrated. We will continue to investigate the relationships between liquid flow, nano-object transport, and ionic current, visualizing the behavior in the nanochannels. Further results will be reported elsewhere in the future.

Recently, various studies related to a novel nanoscale pH sensor [44] or dual-gated Si-nanowire field-effect transistor sensor [45] that beat the Nernst limit (59.1 mV/pH at 298 K) have been reported. Flexible wearable sensors that identify pH values from sweat, urine, tears, and saliva have also been proposed for monitoring health and the early detection of diseases. These kinds of biosensors require high sensitivity near pH 7 because biological cells stop functioning below 6.8 or above 7.8 [46]. Furthermore, several optical fiber pH sensors have also been developed whose wavelength shifts quickly and linearly with proton concentration [47,48]. Compared to the other methods, our method of identifying proton concentration using electrodiffusioosmosis under steady current conditions is unique and enables us to increase the pH sensitivity, which may have advantages in diagnostic analysis. Furthermore, a rapid response comparable to optical measurements should be achieved in the future.

## 4. Conclusions

In the present study, we proposed three types of micro- and nanochannel systems for pH sensors. Based on proton selectivity and electrodiffusioosmosis, ICR effects work well to quantitatively identify proton concentrations. It was also found that separation of the WE, CE, RE, and sample solutions connected with nanochannels was important in the design of micro- and nanochannel systems. In particular, the nanochannel filled with a mixture of PEG and 10 mM KCl solution drastically improved the measurement stability, reproducibility, and the linearity of *V*–pH characteristics. Under galvanostatic current conditions, the slope was 243 mV/pH at 3 nA in the range from pH 1.68 to 10.01, exceeding the Nernst limit of 59.1 mV/pH. The measurement stability was also improved compared to measurements under equilibrium conditions. Although further improvements in the ICR ratios may be required for higher measurement accuracy and reliability, the present conceptual idea is expected to be available for the identification of various ions.

## Figures and Tables

**Figure 1 micromachines-15-00698-f001:**
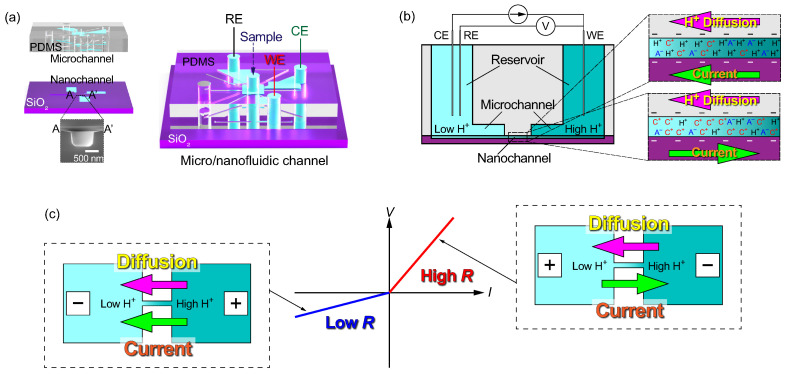
(**a**) Illustration of a microchannel (upper) and nanochannel (lower) formed by bonding them to each other, together with an SEM image. The working, counter, and reference electrodes are placed in reservoirs connected to the sample solution through nanochannels. (**b**) Conceptual diagram of electrodiffusioosmosis in a nanochannel, which is induced by applying a galvanostatic current. (**c**) Schematic diagram of ionic current rectification caused by the difference in the direction of the conduction and diffusion of the ionic current.

**Figure 2 micromachines-15-00698-f002:**
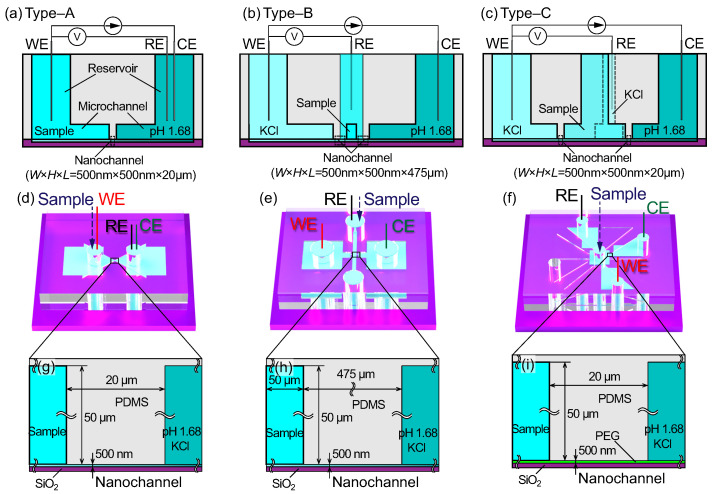
Conceptual diagrams of micro- and nanofluidic channels (**a**) Type A, (**b**) Type B, and (**c**) Type C: cross-sectional views drawn without considering dimensions for convenience. Configurations of the working, counter, and reference electrodes, WE, CE, and RE, respectively, and solutions are schematically depicted. Three-dimensional view of assembled micro- and nanochannels (**d**) Type A, (**e**) Type B, and (**f**) Type C, corresponding to (**a**–**c**), respectively, and focused view of the test sections for (**g**) Type A, (**h**) Type B, and (**i**) Type C and their actual aspect ratios. The test section was designed with the dimensions of width, height, and length: 500 nm, 500 nm, and 20 µm for Type A anc C, and 500 nm, 500 nm, and 475 µm for Type B, respectively. A constant current was applied between the WE and CE, and the potential difference between the electrodes was measured.

**Figure 3 micromachines-15-00698-f003:**
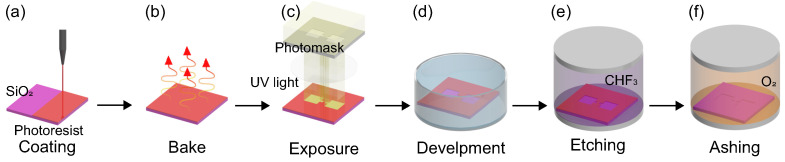
Schematic illustration of the nanochannel fabrication process. (**a**,**b**) A Si/SiO_2_ wafer was coated with photoresist and baked to evaporate the solvent. (**c**) The photoresist was exposed to UV light through a nanochannel photomask using an i-line stepper. (**d**) The photoresist was developed, and the channel pattern was printed on the surface. (**e**) Nanochannels were formed on the SiO_2_ surface using reactive ion etching. (**f**) The residual photoresist was removed by oxygen ashing.

**Figure 4 micromachines-15-00698-f004:**
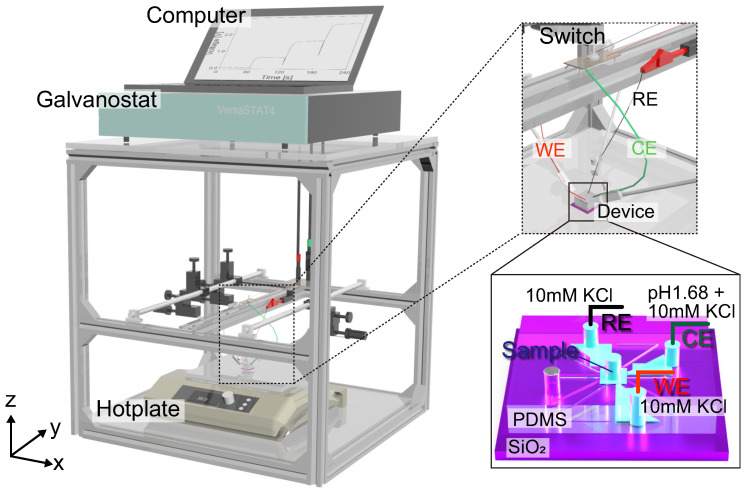
Schematic illustration of the experimental setup: a micro- and nanochannel chip was placed on a hotplate, and the ionic current was measured by a galvanostat using a two- or three-electrode method depending on the channel type. The direction of the galvanostatic current was changed using a mechanical switch. The figure shows a Type C channel, where the WE, CE, RE, and sample solution were in different reservoirs.

**Figure 5 micromachines-15-00698-f005:**
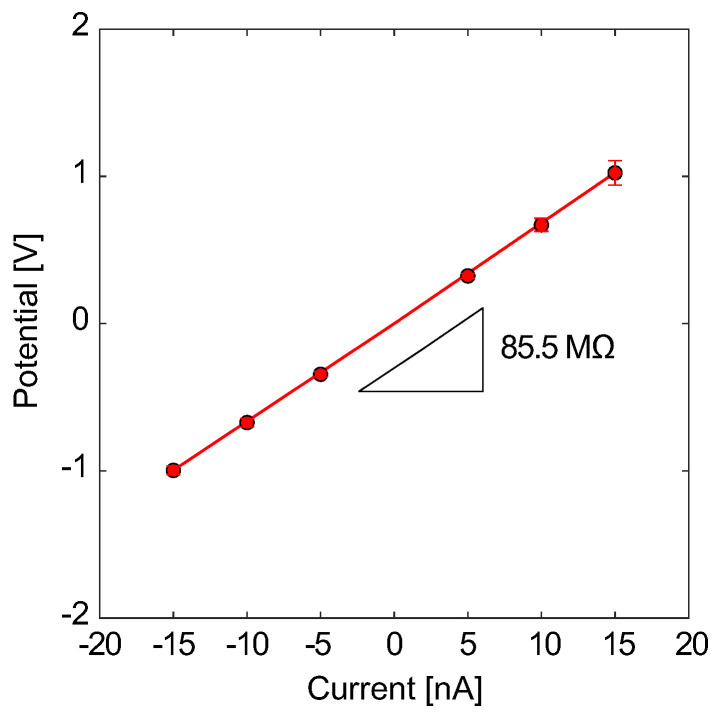
*I*–*V* characteristic for a Type A channel filled with a mixture of 10 mM KCl and pH 1.68 buffer solutions and measured at a temperature of 308 K using the experimental setup shown in Figure 2. The solid line presents the average for N=5 samples.

**Figure 6 micromachines-15-00698-f006:**
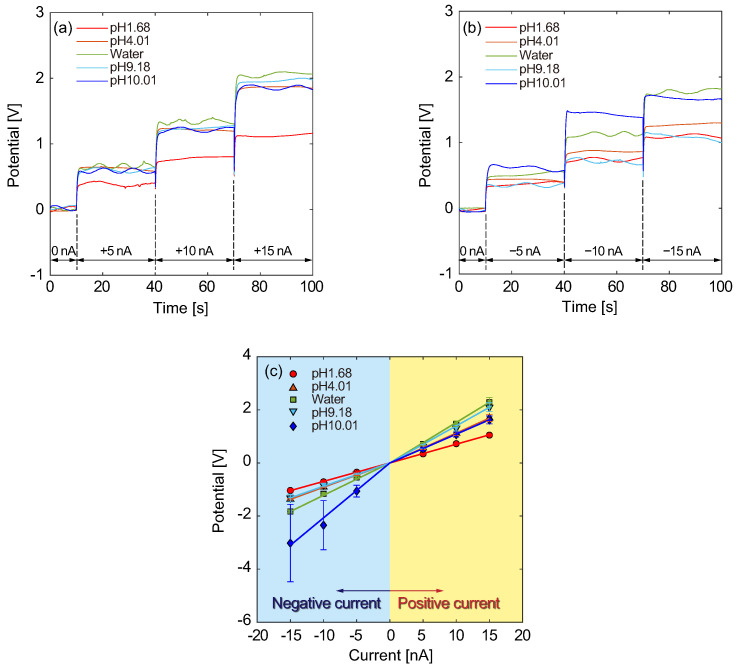
Typical electrical potential responses for the Type A structure for various sample solutions ranging from pH 1.68 to 10.01 for (**a**) positive and (**b**) negative galvanostatic currents. (**c**) *I*–*V* characteristics resulting from (**a**,**b**), where solid lines present the averages for N=5 samples. Galvanostatic currents varying in the range from −15 to 15 nA were maintained for 30 s each and switched at ±5 nA intervals. The temperature was maintained at 308 K.

**Figure 7 micromachines-15-00698-f007:**
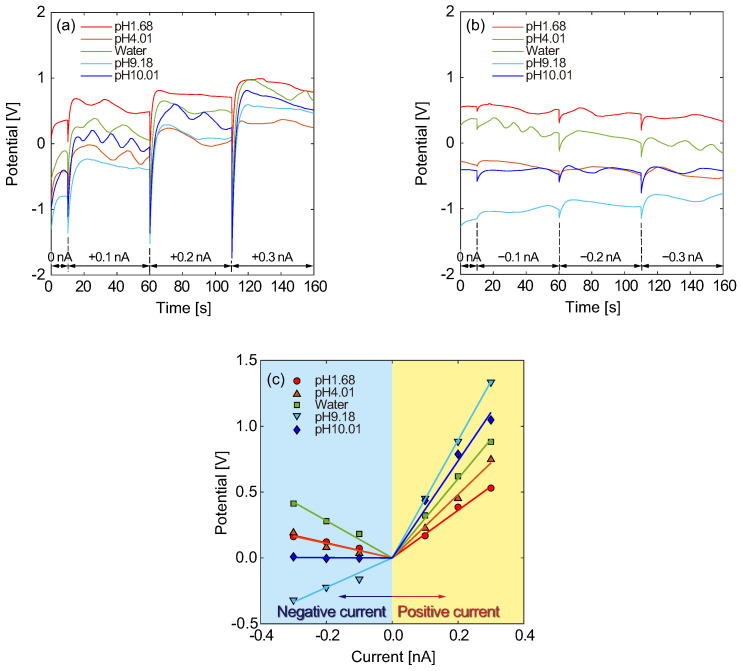
Typical electrical potential responses of a Type B structure for various sample solutions ranging from pH 1.68 to 10.01 for (**a**) positive and (**b**) negative galvanostatic currents. (**c**) *I*–*V* characteristics resulting from (**a**,**b**), where solid lines present the averages for an N=1 sample. Galvanostatic currents varying in the range from −0.3 to 0.3 nA were maintained for 50 s each and switched at ±0.1 nA intervals. The temperature was maintained at 308 K.

**Figure 8 micromachines-15-00698-f008:**
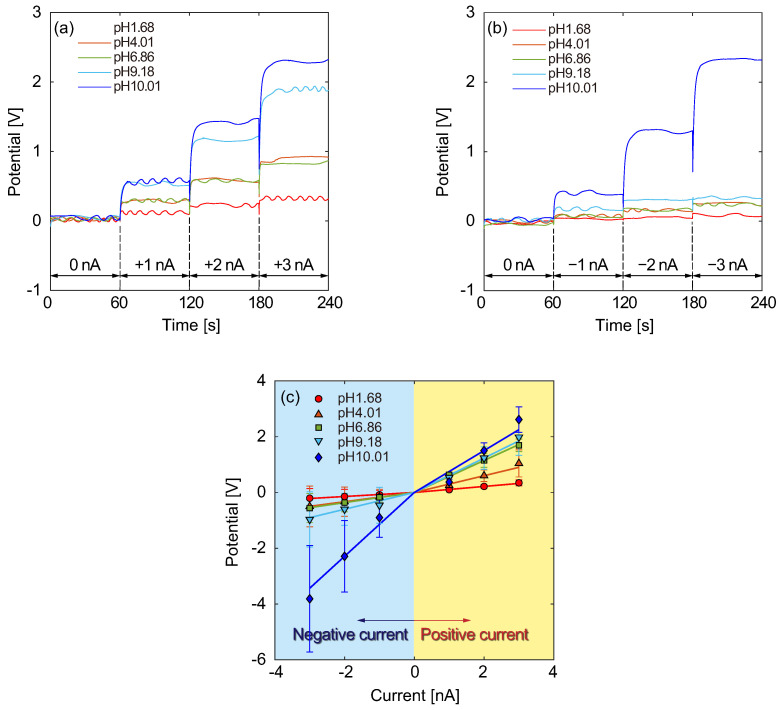
Typical electrical potential responses of the Type C structure for various sample solutions ranging from pH 1.68 to 10.01 for (**a**) positive and (**b**) negative galvanostatic currents. (**c**) *I*–*V* characteristics resulting from (**a**,**b**), where solid lines present the averages for N=13 for pH 1.68, 4.01, and 9.18, N=3 for pH 6.86, and N=7 for pH 10.01. Galvanostatic currents varying in the range from −3 to 3 nA were maintained for 60 s each and switched at ±0.1 nA intervals. The temperature was maintained at 308 K.

**Figure 9 micromachines-15-00698-f009:**
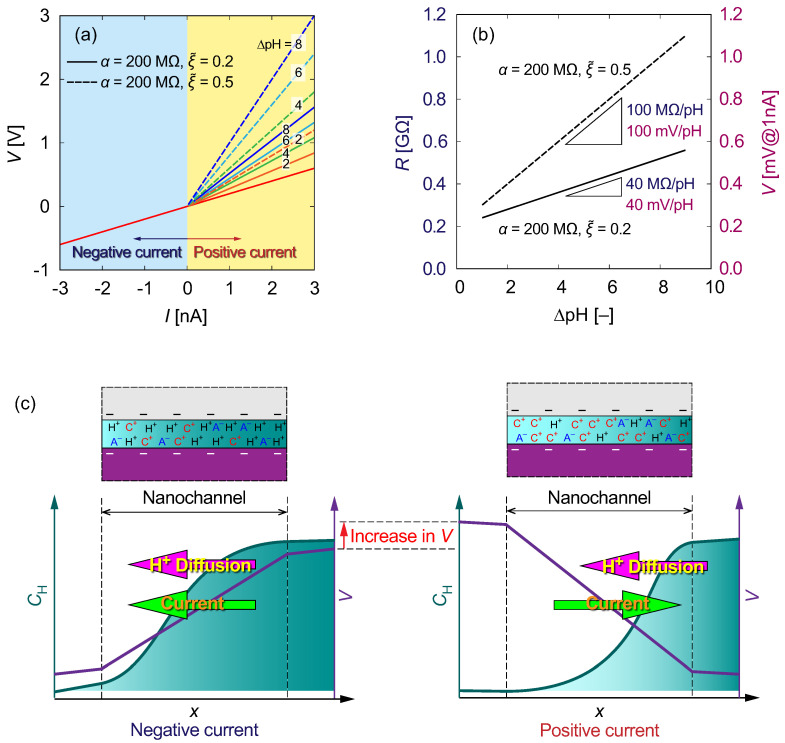
(**a**) *I*–*V* and (**b**) *R*–pH characteristics evaluated using the theoretical model Equations (Equation 5) and (Equation 6) in Appendix A for parameters α=200 MΩ and $\tilde{\xi }=0.2$ (solid line) and 0.5 (dashed line). Multiplying the resistance by the ionic current value, e.g., I=1 nA, converts the slope to a potential difference per pH. (**c**) Schematic illustration of the proton distribution ($C_{H}$) and electrical potential (*V*) in the nanochannel for the positive and negative currents corresponding to (**a**) and the experiments.

**Figure 10 micromachines-15-00698-f010:**
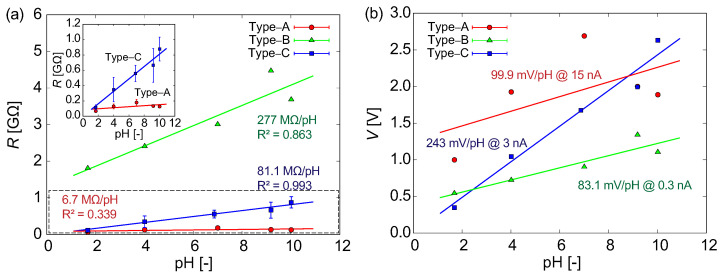
(**a**) *R*–pH characteristics for Type A, B, and C channels as shown in Figure 6, Figure 7, Figure 8, respectively. The solid lines present the theoretical model of Equation (Equation 5) in Appendix A. The slopes are 6.7, 277, and 81.1 MΩ/pH, and the intercepts are 84.2, 1308, and 0 MΩ for Type A, B, and C, respectively. A focused view for Type A and C is shown in the inset. (**b**) *V*–pH characteristics for Type A, B, and C for galvanostatic currents of 15, 3, and 0.3 nA, respectively, which are the maximum currents applied in each experiment. The slopes in (**a**) are also translated to 99.9, 83.1, and 243 mV/pH, respectively.

**Table 1 micromachines-15-00698-t001:** Standard pH sample solutions.

pH	Name	Composition
1.68	Oxalate	49.61 mmol/L KH_3_(C_2_O_4_)_2_·2H_2_O
4.01	Phthalate	49.55 mmol/L C_6_H_4_(COOK)(COOH)
6.86	Phosphate	25 mmol/L KH_2_PO_4_ and 25 mmol/L Na_2_HPO_4_
9.01	Tetraborate	9.964 mmol/L Na_2_B_4_O_7_·10H_2_O
10.01	Carbonate	24.90 mmol/L NaHCO_3_ and 24.91 mmol/L Na_2_CO_3_

## Data Availability

The original contributions presented in this study are included in the article, further inquiries can be directed to the corresponding author.

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
