# Peer review of "Micro- and Nanofluidic pH Sensors Based on Electrodiffusioosmosis"

_micromachines, 2024, doi:10.3390/mi15060698_

Round 1
Reviewer 1 Report (New Reviewer)
Comments and Suggestions for Authors
This paper presents the nanofluidic devices focusing on ionic current rectification (ICR) for pH sensing. Electrodiffusioosmosis induced in nanochannels enhances pH sensor performance, with potential applications across various ions. The design approach is fine. Some concerns need to be addressed in the revision.
1. To provide a thorough analysis of micro pH sensors, authors should expand their comparison to encompass a broader range of sensing mechanisms beyond the ones mentioned. This includes but is not limited to potentiometric sensors, optical sensors, electrochemical sensors, and surface plasmon resonance sensors. By incorporating these additional types of sensors, the authors can offer a more comprehensive evaluation of micro pH sensors' strengths, limitations, and potential applications relative to other sensing technologies. The following papers should be discussed. 10.1039/D0RA00016G, 10.1364/OE.511190, 10.3390/s21124218
2. How does using nanochannels in this study enhance the sensitivity and accuracy of the micro pH sensors compared to conventional micro-scale pH sensors? More clarification is expected.
3. Can authors explain how the design variations between Type-A, Type-B, and Type-C nanochannels influence the performance and applicability of the micro pH sensors, particularly in terms of sample handling and electrode configuration?
4. Considering the improvements seen in Type-C compared to Type-A and Type-B, could the authors discuss potential limitations or challenges associated with the fabrication or implementation of Type-C micro- and nanochannels, particularly regarding scalability and reproducibility for practical applications?
Comments on the Quality of English Language
The current version contains some typos and grammatical errors throughout. A thorough proofreading is essential to address these issues and ensure the manuscript's clarity and professionalism.
Author Response
Response to Reviewer 1
The authors appreciate the reviewer for your time to review our manuscript. We sincerely received your comments and carefully revised the manuscript according to your instructions. One-to-one responses to the comments were provided as below. The corrections are highlighted with blue text in the manuscript. We hope that the revisions sufficiently satisfy the criticisms and meet the quality of acceptance.
1. To provide a thorough analysis of micro pH sensors, authors should expand their comparison to encompass a broader range of sensing mechanisms beyond the ones mentioned. This includes but is not limited to potentiometric sensors, optical sensors, electrochemical sensors, and surface plasmon resonance sensors. By incorporating these additional types of sensors, the authors can offer a more comprehensive evaluation of micro pH sensors' strengths, limitations, and potential applications relative to other sensing technologies. The following papers should be discussed. 10.1039/D0RA00016G,10.1364/OE.511190, 10.3390/s21124218
Response: The authors appreciate the reviewer’s important instruction. Referring to the articles introduced by the reviewer, we compared the present sensor with previously reported various types and emphasized the strong points and limitations of our method at L. 373‒382.
2. How does using nanochannels in this study enhance the sensitivity and accuracy of the micro pH sensors compared to conventional micro-scale pH sensors? More clarification is expected.
Response: In the present method, electrodiffusioosmosis associated with ICR effects through nanochannels effectively work to identify proton concentrations in sample solutions. This method improves measurement stability due to the steady current conditions compared to the measurement methods based on equilibrium states, which are affected by thermal fluctuations. Furthermore, various ion species as well as proton are expected to be identify by the present method by changing the reference solutions for the probes, which does not require chemical modifications of nanochannels. This point was additionally emphasized in the Abstract and Conclusions sections.
3. Can authors explain how the design variations between Type-A, Type-B, and Type-C nanochannels influence the performance and applicability of the micro pH sensors, particularly in terms of sample handling and electrode configuration?
Response: The authors appreciate the reviewer’s comment. The variety of micro- and nanochannels were historically developed to improve the measurement accuracy and ease of sample handling, as you mentioned. We mainly focused on the configuration of RE and sample solutions and found that the RE and sample solutions should be isolated from the other electrodes. In Type‒A, both RE and CE were settled into the reference solution, and then, the ICR ratio was small and did not identify pH values well. In Type‒B, the three electrodes were isolated from each other, and the RE was settled into the sample solution. The ICR ratio was improved, but the I‒V characteristics did not distinguish alkaline solutions. In Type‒C, the sample solution was fully isolated from the three electrodes and was connected to them via nanochannels. As a result, the pH value of alkaline solutions was clearly identified. Differences between the three types are additionally described at L. 103‒110.
4. Considering the improvements seen in Type-C compared to Type-A and Type-B, could the authors discuss potential limitations or challenges associated with the fabrication or implementation of Type-C micro- and nanochannels, particularly regarding scalability and reproducibility for practical applications?
Response: The authors appreciate your important comment. Type-C channel was the best in the three types of channels. On the other hand, the ICR ratio should be improved to clarify the electrodiffusioosmosis through the negatively charged nanochannels, which is the fundamental principle of the present method. The difference in conductivities depending on the ionic current direction relative to the concentration gradient is an important factor to identify various ion species as well as proton. Further improvement of the ICR ratio is an important issue in the future work. This point was emphasized at L. 359‒372.
Additionally, grammatical errors and typos were corrected by native speakers, and minor corrections were made without highlighting.
--
Reviewer 2 Report (New Reviewer)
Comments and Suggestions for Authors
The manuscript deserves publication after minor revisions:
1) The abstract could be rewritten considering the novelty and originality as well as the most important numerical results.
2) The introduction could be reduced. The schemes described here could be reported in the materials and methods sections or in the discussion of the results.
3) Several information reported in the figure 2 could be described in the materials section.
4) All figure captions should include the most important experimental information.
5) hydrodynamic conditions should be determined by the electrochemical measurements to classify all designs.
6) Model written by using mathematical equations or Excel, it should be reported here should be included as supplementary information.
7) Conclusions could be improved.
Author Response
Response to Reviewer 2
The authors appreciate the reviewer for your time to review our manuscript. We sincerely received your comments and carefully revised the manuscript according to your instructions. One-to-one responses to the comments were provided as below. The corrections are highlighted with blue text in the manuscript. We hope that the revisions sufficiently satisfy the criticisms and meet the quality of acceptance.
1. The abstract could be rewritten considering the novelty and originality as well as the most important numerical results.
Response: The authors appreciate the reviewer’s comment. The originality of the present study and new findings were additionally emphasized in the Abstract section.
2. The introduction could be reduced. The schemes described here could be reported in the materials and methods sections or in the discussion of the results.
Response: According to the reviewer’s instruction, the volume of introductory part was reduced. Descriptions related to the methods and results were reduced or moved to the methodology part and the results and discussion, respectively.
3. Several information reported in the figure 2 could be described in the materials section.
Response: Detailed explanation about Figure 2 were additionally written in the Materials and Methods: L. 97‒98, 101‒102, 103‒110.
4. All figure captions should include the most important experimental information.
Response: According to the reviewer’s instruction, detailed information of the experiments was additionally described in all figure captions.
5. hydrodynamic conditions should be determined by the electrochemical measurements to classify all designs.
Response: The authors appreciate the reviewer’s critical comment. We certainly agree that the hydrodynamic conditions are effective on the ion transport phenomena. However, there is a difficulty to detect detailed phenomena that occur in the nanochannel in the present system. Therefore, the design of nanochannels is uniquely determined in each type, which is 500 nm wide, 500 nm heigh, and 20 μm long, based on our previous experiences. In the experiments, the heads of solutions in each reservoir were initially equal each other, and we expected that the effect of hydrostatic pressure on flows in the channels was weak. In steady current conditions, EOFs are driven by external electric fields and they will be included in the conductivity. Differences in the designs of reservoirs and microchannels may cause the measurement errors and reproducibility, and they should be improved for higher precisions. Among the three channel types, Type‒C was the most stable due to the separation of each reservoir by nanochannels, and this may be caused by the suppression of hydrodynamic effects. We have continued to visualize liquid flows and transport of nanoobjects in nanochannels in other studies and will report elsewhere in the future. This point was additionally described at the last paragraph in the results and discussion. This point was emphasized at L. 363‒368.
6. Model written by using mathematical equations or Excel, it should be reported here should be included as supplementary information.
Response: The mathematical model to explain the electrodiffusioosmosis in the experimental system was moved to Appendix. Numerical results were only remained in the results and discussion.
7. Conclusions could be improved.
Response: According to the reviewer’s comment. Conclusions were improved, emphasizing new findings and improvements of previous studies.
Additionally, grammatical errors and typos were corrected by native speakers, and minor corrections were made without highlighting.
--
Round 2
Reviewer 1 Report (New Reviewer)
Comments and Suggestions for Authors
The authors have addressed my concerns. I suggest accepting this paper for publication.
This manuscript is a resubmission of an earlier submission. The following is a list of the peer review reports and author responses from that submission.
Round 1
Reviewer 1 Report
Comments and Suggestions for Authors
This paper primarily focuses on the utilization of a mixture of polyethylene glycol and potassium chloride solution to fill the nanochannels, and the successful induction of electrodiffusioosmosis through the phenomenon of ionic current rectification (ICR). A pH sensor based on this technology was developed. The objective was to leverage the ion selectivity and rectification of ionic currents in the nanochannels for measuring the concentration of the sample. This innovative approach exhibits significant improvements in stability and measurement precision compared to traditional sensors, demonstrating tremendous potential.The research content of the paper is interesting and engaging, demonstrating a certain level of innovation. However, there are some issues that need clarification:
(1) Why were these three different types of nanochannels chosen for comparison, and what is the basis for this selection?
(2) The article primarily focuses on analyzing and comparing the linear relationship of the sensors, with limited discussion of data trends, deviations, or abnormal results.
(3) The article mentions an improvement in measurement accuracy but fails to provide valid data for verification.
(4) The article only discusses the performance of the proposed sensor without comparing it to existing sensor technologies, making it difficult to assess the method's advantages in terms of performance and applications.
Author Response
Response to Reviewer 1
The authors appreciate the reviewer for your time to review our manuscript. We sincerely received your comments and carefully revised the manuscript according to your instructions. One-to-one responses to the comments were provided as below. We hope that the revisions sufficiently satisfy the criticisms and meet the quality of acceptance.
(1) Why were these three different types of nanochannels chosen for comparison, and what is the basis for this selection?
The authors appreciate the reviewer’s comment. In this study, the authors investigated the effects of the structures of micro- and nanochannels on the current–voltage characteristics of pH sample solutions. Furthermore, improvements by the three-electrode method, in which the reference electrode was separated from the counter electrode, were clarified by comparing with the two-electrode method. In Type–A, two reservoirs were separated and connected by a nanochannel, and the current–voltage characteristics was measured by the two-electrode method. In Type–B, two reservoirs for the working and counter electrodes were separated from the reference electrode that was settled into a reservoir of a microchannel filled with sample solutions, and they were connected with nanochannels. The current–voltage characteristics was measured by the three-electrode method. In Type–C, three reservoirs for each electrode and sample solution were separated by nanochannels filled with PEG solution, and the current–voltage characteristics was measured by the three-electrode method. As a result, it was found that each solution should be stable in the reservoirs and that the high impedance of nanochannels that bridge each reservoir was effective on the measurement stability and reproducibility. This point is highlighted at L. 78–82, P.3.
(2) The article primarily focuses on analyzing and comparing the linear relationship of the sensors, with limited discussion of data trends, deviations, or abnormal results.
The experimental results from the three types were discussed with respect to electrodiffusioosmosis and further discussed theoretically using schematic diagrams as shown in Figures 9 and 10. Discussions about this point were additionally described at L.379 and 385–387, P.13 and L. 401–14, P. 14.
(3) The article mentions an improvement in measurement accuracy but fails to provide valid data for verification.
In the manuscript, it was emphasized that the averages were determined with linearity and small deviations by the improvement in measurement accuracy. In this study, the pH values of sample solutions (standard solutions) were verified in advance by using a pH meter (LAQUAtwin, Horiba, Kyoto, Japan). On the other hand, the present method cannot determine the absolute pH values and requires providing a calibration curve. In other words, this study is the first step to determine the calibration curve using the micro- and nanofluidic channel devices. The usage of measurement accuracy may be misleading. Descriptions about the measurement accuracy were explicitly replaced by the linearity as a function of pH values and small deviations for the averages. A subsection about preparation for sample solutions was added with Table 1 to explain descriptions above at L. 148–161, P. 5.
(4) The article only discusses the performance of the proposed sensor without comparing it to existing sensor technologies, making it difficult to assess the method's advantages in terms of performance and applications.
Conventional pH analysis methods depend on the electrical potential difference at equilibrium caused by the difference in proton concentrations, where two solutions are separated with a glass membrane that has proton selectivity. In an equilibrium state, the slope of electrical potential difference is determined by the Nernst equation that results in 59.1 mV/pH at 298.15 K. This value is theoretically and experimentally known to be correct and is used as a reference. Comparing with this value, the present experimental results were discussed. In previous studies, Refs. 33 and 34, we also demonstrated the linearity of electrical potential differences as a function of pH values, using double- and triple-barreled glass microelectrodes. We emphasized that the linearity for pH values were also confirmed using nanochannels fabricated on SiO2 thin film and that the slope of electrical potential difference larger than 59.1 mV at 298 K (61.1 mV at 308 K) was achieved by the constant current measurement. Descriptions about recent various methods compared to our method were additionally written in the discussion at L.409–414, P.14.
--
Reviewer 2 Report
Comments and Suggestions for Authors
This manuscript presents an interesting work. There are some points that are not clear.
1、 The currents applied to Type A are different from those applied to Type B and C. How to select the current? Dose the current effect the sensitivity or stability?
2、 What are the materials of WE, CE and RE? Are they Ag/AgCl wires?Ag/AgCl can be used as RE. I don't think Ag/AgCl can be used as WE and CE. Please give some explanation.
3、 In type-C, is there an empty reservoir? Please give some explanation.
4、 As described in section 3.5 and shown in Fig.10, the slope of Type–B is 417 mV/pH. However, I think this result can’t be obtained from Fig.7. Please check it and give some explanation.
Author Response
Response to Reviewer 2
The authors appreciate the reviewer for your time to review our manuscript. We sincerely received your comments and carefully revised the manuscript according to your instructions. One-to-one responses to the comments were provided as below. We hope that the revisions sufficiently satisfy the criticisms and meet the quality of acceptance.
- The currents applied to Type A are different from those applied to Type B and C. How to select the current? Dose the current effect the sensitivity or stability?
We appreciate the reviewer for the critical comment. An important point of this method is to find the preferable condition in which diffusion and electrokinetic transport of ions are balanced to maintain a steady state. From the viewpoint of electrodiffusioosmosis, it is suggested that an applied current should be weak enough to perturb an equilibrium state even slightly. On the other hand, the electrical measurements tend to be disturbed by noise in excessively weak current conditions. To satisfy both requirements above, the applied current value has to be optimally determined to maintain the stability and reproducibility of measurements. Therefore, the preferred current values vary depending on the channel designs and concentrations. For the information, the electrical potential measurement was limited to a range within ±10 V, but this limitation was not a problem for each type. This point was additionally described at L. 188–190, P. 6.
- What are the materials of WE, CE and RE? Are they Ag/AgCl wires? Ag/AgCl can be used as RE. I don’t think Ag/AgCl can be used as WE and CE. Please give some explanation.
All electrode materials used in the present experiments are Ag–AgCl wires. In the present study, a galvanostat was used to perform constant current measurements, which was served as a power source and amplifier. To maintain constant current conditions, redox reactions of Ag/AgCl were used: Ag + Cl− → AgCl + e− at anode, AgCl + e− → Ag + Cl− at cathode. In the experiments, the WE and CE were merely used for the positive electrode (anode) and the negative electrode (cathode), respectively. To perform the potentiometry in liquids, the three electrodes are required, although the names WE, CE, and RE may be specific to electrochemical measurements. This point was additionally explained at L. 191–195, P. 6
- In type-C, is there an empty reservoir? Please give some explanation.
Although type–C is designed to be applicable to multi-electrode measurements in the future, some reservoirs have remained empty in this study. This point was additionally described in the main text at L. 186–187, P. 6.
- As described in section 3.5 and shown in Fig.10, the slope of Type–B is 417 mV/pH. However, I think this result can’t be obtained from Fig.7. Please check it and give some explanation.
We appreciate the reviewer’s comments. The errors you pointed out were corrected and Figure 10 was improved to show the slopes of resistance and potential difference as a function of pH values.
--
Reviewer 3 Report
Comments and Suggestions for Authors
The paper on Micro- and Nanofuidic pH Sensors… is missing many important information on the experimental systems and measuring procedure that in present version cannot be considered a scientific paper which should allow the experiment to be reproduced.
1. What is the experimental setup? Figure 2 is misleading. There is a channel of 20 micrometers in Fig.2.g but in text there is nanochannel of 500x500nm. There are reservoirs of 1x1 mm and the depth of 500nm ???. There are no electrodes in a system. I would strongly recommend drawing a picture of cross section of the apparatus observed from the side (and not from the top). The drawing should be out of scale with dimensions written in the drawing. There should be electrodes and the level of solutions applied.
2. The composition of the media should be described. The buffer is not sufficient information on the chemicals which are in a system. Since the adsorption on the nanochannel surface might be an important factor in experiment why authors used buffers instead of HCl and KOH addition? How authors checked the pH of the medium used during and after the experiment.
3. The description of electrodes as Ag-AgCl wires are not sufficient. What were the sizes of electrodes? How they were made? Was the difference between working and reference electrodes? How they were imbedded into the reservoir of the depth of 0.5 micrometer?
4. Detailed description of experimental procedure is required. How the apparatus was filled with solutions? Capillary forces will fill the narrow part of apparatus from the side of first solution added. How the solutions in both reservoirs were leveled to prevent hydraulic flow? How apparatus was washed before changing the medium? Or how the media were replaced? Does the current clamp mode was introduced? Was there a sweep of current values during the experiment?
5. High surface conductivitties of hydrogen ions were described 100+ years ago. It would be nice if authors will just mention this fact giving one or two references mentioning their predecessors.
Comments on the Quality of English LanguageEnglish is not a problem. Unclear text is.
Author Response
Response to Reviewer 3
The authors appreciate the reviewer for your time to review our manuscript. We sincerely received your comments and carefully revised the manuscript according to your instructions. One-to-one responses to the comments were provided as below. We hope that the revisions sufficiently satisfy the criticisms and meet the quality of acceptance.
- What is the experimental setup? Figure 2 is misleading. There is a channel of 20 micrometers in Fig.2.g but in text there is nanochannel of 500x500nm. There are reservoirs of 1x1 mm and the depth of 500 nm???. There are no electrodes in a system. I would strongly recommend drawing a picture of cross section of the apparatus observed from the side (and not from the top). The drawing should be out of scale with dimensions written in the drawing. There should be electrodes and the level of solutions applied.
We appreciate the reviewer’s comment. According to the reviewer’s instruction, cross-sectional views (side views) of micro- and nanochannels were drawn in Figure 2. The positions of electrodes were also depicted.
- The composition of the media should be described. The buffer is not sufficient information on the chemicals which are in a system. Since the adsorption on the nanochannel surface might be an important factor in experiment why authors used buffers instead of HCl and KOH addition? How authors checked the pH of the medium used during and after the experiment.
According to the reviewer’s comment, details about pH solutions used in the experiments were summarized in Table 1. The pH values of sample solutions, which were commercially available pH buffer solutions (standard solutions), were confirmed by using a conventional pH meter before experiments. It is difficult to measure the pH values after experiments because of the small quantity of liquids. Therefore, the pH buffer solutions were used to maintain the constant pH values during the experiments. In our future plan, a variety of solutions will be measured using the present system. On the other hand, the solutions, which stably maintain the pH values, should be used for the test as a first step, as the reviewer is concerned. Experiments using prepared HCl or KOH are the next step. A subsection of preparation for sample solutions was added as subsection 2.3 in P. 5.
- The description of electrodes as Ag-AgCl wires are not sufficient. What were the sizes of electrodes? How they were made? Was the difference between working and reference electrodes? How they were imbedded into the reservoir of the depth of 0.5 micrometer?
We appreciate the reviewer’s critical comment. Details about the Ag—AgCl wires, difference between the working and reference electrodes, and how to contact the Ag—AgCl wires with solutions were additionally explained in the main text with some figures. A Ag—AgCl wire was made from a Ag wire with a diameter of 0.3 μm, which was immersed in sodium hypochlorite aqueous solutions until the surface color changed from silver to black. The reference electrode has a role to determine the reference level of electrical potential, and the electrical potential of the working electrode was measured relative to the reference electrode when a constant ionic current is maintained between the working and counter electrodes. The nanochannels fabricated on a SiO2 substrate were aligned and sealed with the microchannels printed on a PDMS substrate. An approximately 10 mm thick PDMS substrate with through-holes punched using a biopsy punch was used as liquid reservoirs to provide solutions to the nanochannels. Explanations above additionally described at L.191–198, P.6.
- Detailed description of experimental procedure is required. How the apparatus was filled with solutions? Capillary forces will fill the narrow part of apparatus from the side of first solution added. How the solutions in both reservoirs were leveled to prevent hydraulic flow? How apparatus was washed before changing the medium? Or how the media were replaced? Does the current clamp mode was introduced? Was there a sweep of current values during the experiment?
According to the reviewer’s comments, details of experimental procedures were additionally described as follow and in the main text: at L. 168–175, P. 6, where a constant current is clamped for 20 s (Type–A and B) or 60 s (Type–C) to measure a potential difference at a steady state and is sequentially increased. The current direction is inverted by changing the polarization of the electrodes using a mechanical switch. The resolution of applied current is 20 pA and the measurement frequency is set to 10 Hz. An electrical potential difference at a steady state is averaged over the measurement time within 2σ, where σ is the standard deviation. During the measurement, a micro- and nanofluidic channel is fixed on a hot plate set at 308 K by applying thermal grease on the back side; at L.199–210, P.6, The liquid reservoir made by a through hole in the PDMS substrate is approximately 10 mm high and is fully filled with solutions. The solutions are injected into each reservoir using syringes, and the micro- and nanochannel device is degassed in a vacuum chamber to fill the nanochannels with solutions. The filled channels can be confirmed optically using a microscope and electrically by conduction. Furthermore, by additionally injecting the solutions, the reservoirs are fully filled with liquids. Especially, sample solutions are sequentially replaced by a sample with a lower pH. In the sequence, a sample solution is replaced after co-washing using the next sample. In the present experiments, the head of each reservoir is not controlled and the effect on the I–V characteristics may be unknown. Although the visualization of nanochannels will be reported in the near future, this study empirically assumes that small differences in the heads do not significantly affect the experimental results.
- High surface conductivities of hydrogen ions were described 100+ years ago. It would be nice if authors will just mention this fact giving one or two references mentioning their predecessors.
According to the reviewer’s suggestion, we additionally cited three papers with respect to the protons’ abnormal conduction mechanism in water and silica surfaces [J. D. Bernal and R. H. Fowler, A theory of water and ionic solution, with particular reference to hydrogen and hydroxyl ions, J. Chem. Phys. 1 (1933), pp. 515—548; J. H. Anderson and G. A. Parks, The electrical conductivity of silica Gel in the presence of adsorbed water, J. Phys. Chem. 72 (1968), pp. 3662—3668; N. Agmon, The Grotthuss mechanism, Chem. Phys. Lett. 244 (1995), pp. 456—462.] in the introduction at L. 62 an 63, P. 2.
--
Round 2
Reviewer 1 Report
Comments and Suggestions for Authors
The manuscript has been improved.
Author Response
The authors appreciate the reviewer's time and contributions.
Reviewer 3 Report
Comments and Suggestions for Authors
I am afraid that improved paper is still not suitable for publication.
From the Figure 1 one cannot make up what the cross section of the system is. Missing dimensions is serious fault. In the paper four electrodes are described in the figures only three are used. Authors are unaware that they are probably measuring different potential than they think they do. According to the paper silver electrodes had diameter of 0.3 micrometers. To operate such electrodes which cannot be visible by naked eye special holders must be applied. In the paper the depth of electrode well is 0.5 micrometer so how the electrode is imbedded into the solution.
Comments on the Quality of English LanguageThe language is not the problem of the paper